# Simple Rules for Complex Near-Glass-Transition Phenomena in Medium-Sized Schiff Bases

**DOI:** 10.3390/ijms23095185

**Published:** 2022-05-06

**Authors:** Andrzej Nowok, Wioleta Cieślik, Joanna Grelska, Karolina Jurkiewicz, Natalina Makieieva, Teobald Kupka, José Alemán, Robert Musioł, Sebastian Pawlus

**Affiliations:** 1Department of Experimental Physics, Wrocław University of Science and Technology, Wybrzeże Stanisława Wyspiańskiego 27, 50-370 Wrocław, Poland; andrzej.nowok@pwr.edu.pl; 2Faculty of Science and Technology, Institute of Chemistry, University of Silesia in Katowice, 75 Pułku Piechoty 1A, 41-500 Chorzów, Poland; wioleta.cieslik@us.edu.pl; 3Faculty of Science and Technology, August Chełkowski Institute of Physics, University of Silesia in Katowice, 75 Pułku Piechoty 1, 41-500 Chorzów, Poland; joanna.grelska@us.edu.pl (J.G.); karolina.jurkiewicz@us.edu.pl (K.J.); sebastian.pawlus@us.edu.pl (S.P.); 4Silesian Center for Education and Interdisciplinary Research, 75 Pułku Piechoty 1A, 41-500 Chorzów, Poland; 5Department of Chemistry, Opole University, Oleska Street 48, 45-052 Opole, Poland; makieievium@gmail.com (N.M.); teobaldk@gmail.com (T.K.); 6Department of Organic Chemistry, Universidad Autónoma de Madrid, Calle Francisco Tomás y Valiente, 7, Cantoblanco, 28049 Madrid, Spain; jose.aleman@uam.es; 7Institute for Advanced Research in Chemical Sciences (IAdChem), Universidad Autónoma de Madrid, 28049 Madrid, Spain

**Keywords:** glass transition, supercooled liquid, Schiff bases, dielectric spectroscopy, bifurcated hydrogen bonds, molecular mobility, self-organization

## Abstract

Glass-forming ability is one of the most desired properties of organic compounds dedicated to optoelectronic applications. Therefore, finding general structure–property relationships and other rules governing vitrification and related near-glass-transition phenomena is a burning issue for numerous compound families, such as Schiff bases. Hence, we employ differential scanning calorimetry, broadband dielectric spectroscopy, X-ray diffraction and quantum density functional theory calculations to investigate near-glass-transition phenomena, as well as ambient- and high-pressure molecular dynamics for two structurally related Schiff bases belonging to the family of glycine imino esters. Firstly, the surprising great stability of the supercooled liquid phase is shown for these compounds, also under high-pressure conditions. Secondly, atypical self-organization via bifurcated hydrogen bonds into lasting centrosymmetric dimers is proven. Finally, by comparing the obtained results with the previous report, some general rules that govern ambient- and high-pressure molecular dynamics and near-glass transition phenomena are derived for the family of glycine imino esters. Particularly, we derive a mathematical formula to predict and tune their glass transition temperature (*T_g_*) and its pressure coefficient (d*T_g_*/d*p*). We also show that, surprisingly, despite the presence of intra- and intermolecular hydrogen bonds, van der Waals and dipole–dipole interactions are the main forces governing molecular dynamics and dielectric properties of glycine imino esters.

## 1. Introduction

The characterization of near-glass-transition phenomena and the vitrification process itself have constituted an object of intense research for many years. However, despite numerous studies, a deep understanding of the glassy state along with structure–property relationships still remains an open challenge [1,2,3,4]. The importance of this issue is apparent in material sciences, in which tunability of physicochemical properties in organic compounds has become a burning issue. For optoelectronic purposes, it can be achieved by structure-based molecular design and selecting crucial chemical motifs. One of them is the imine C=N double bond, characteristic of the family of Schiff bases [5,6,7,8].

Since being first synthesized by Hugo Schiff, Schiff bases have experienced a research explosion and have become valuable materials for synthesis, medicine, biotechnology, photovoltaics, optoelectronics or electronics [5,6,7,8,9,10,11,12,13]. Electronic applications originate from their π-conjugated, hole-transporting semiconducting properties [14]. However, particular attention has recently been attracted by their o-hydroxyphenyl derivatives due to fluorescence, thermo-, photo- and solvatochromism, as well as the formation of intra-intermolecular bifurcated hydrogen bonds (BHBs) [15,16,17,18,19,20,21]. In general, these BHBs are three-centered hydrogen bonds with one hydrogen atom interacting simultaneously with two proton-acceptor moieties (nitrogen and oxygen) [22,23]. Their most atypical features are particular high strength (short length), as well as environment-dependent and resonance-assisted proton-transfer equilibrium in the intramolecular N^…^H^…^O branch [24,25,26]. Consequently, *o*-hydroxyphenyl Schiff bases can be switched between enol-imine, keto-enamine and zwitterionic forms, which makes them interesting materials for sensors, optical switches or memory devices [20]. On top of that, some compounds with the imine C=N bond possess glass-forming properties with high glass transition temperatures (*T_g_* > 298 K) [6,7,8,21]. Unfortunately, only a few of them were investigated by means of broadband dielectric spectroscopy and only under ambient-pressure conditions, so the knowledge about their dielectric response, molecular dynamics and near-glass-transition phenomena is highly limited [21]. The current consensus is that even sterically hindered Schiff bases can organize themselves into dimers via BHBs in supercooled liquid (just like in crystals). Due to their centrosymmetric architecture, the dielectric response of supercooled Schiff bases is not dominated by the Debye process, typical for other self-organizing H-bonded systems (e.g., alcohols, amines) [21]. Apart from this common denominator, physicochemical properties of glass-forming compounds with imine C=N bond seem to be highly differentiated, according to previous reports [6,7,8,21]. First, their *T_g_* values range from 282 up to 563 K. Secondly, Schiff bases span between low-molecular-weight and sizeable systems. Meanwhile, these groups were reported to have completely different molecular dynamics and dielectric response [27,28,29]. For example, anticorrelation between dielectric strength and frequency dispersion of α-relaxation (established for low-molecular-weight compounds) is violated in sizeable systems [27]. Moreover, the relaxation time in the limit of infinitely high temperatures becomes incomparably longer, substantially exceeding even the typical phonon-like time scale (10^−14^ s) [29]. Naturally, the question arises: are there any general rules that govern complex near-glass-transition phenomena in Schiff bases under ambient- and high-pressure conditions?

Motivated by the shortcomings, we screen the physicochemical properties of numerous glass-forming Schiff bases in search of simple mathematical formulas describing their structure–dynamics relationships. Particular emphasis is put on the family of glycine imino esters (promising for the synthesis of optically active compounds [30,31,32,33]), two medium-sized representatives, (**1** and **2**) of which are investigated herein. Finally, based on quantum density functional theory (DFT) computation, Fourier transform infrared (FTIR) and broadband dielectric (BDS) spectroscopy, as well as X-ray diffraction (XRD), we show that ambient- and high-pressure near-glass-transition behavior of this Schiff bases family is completely different from other H-bonded self-organizing systems.

In this article, we frequently use some symbols and abbreviations apart from those connected with the names of experimental techniques. For the sake of simplicity, we list them in alphabetical order and provide their explanation: *α*-relaxation—structural relaxation, *β_KWW_*—fractional exponent in the Kohlrausch–Williams–Watts function, Δ*ε*—dielectric increment, Δ*V*—activation volume, *ε*″—imaginary part of complex dielectric permittivity, *τ_α_*—structural relaxation time, *τ_β_*—relaxation time of the β-process, BHBs—bifurcated hydrogen bonds, dc-conductivity—direct current conductivity, d*T_g_*/d*p*—pressure coefficient of the glass transition temperature, *E_a_*—activation energy, *E_g_*—calculated energy barrier for conformational change, H bond—hydrogen bond, HN function—Havriliak–Negami function, JG process—Johari–Goldstein process, *k*—Boltzmann constant, *M*—molar mass, *m_p_*—steepness index, *p*—pressure, *T*—temperature, *T_g_*—glass transition temperature, VFTH equation—Vogel–Fulcher–Tamman–Hesse equation.

## 2. Results and Discussion

Schiff bases **1** and **2** are medium-sized and sterically hindered compounds belonging to the subgroup of glycine imino esters. Both compounds differ only in one substituent (see Figure 1a) and, consequently, molar mass (398 and 444 g∙mol^−1^ for analogs **1** and **2**, respectively).

The small structural difference between compounds **1** and **2** exerts a huge impact on their physicochemical properties. As revealed by calorimetric DSC measurements performed with a rate of 10 K∙min^−1^, compound **1** undergoes vitrification at ~264 K, i.e., below room temperature (Figure 1b). In turn, compound **2** (with higher molar mass) forms stable glass phase at 298 K. Thermal anomaly, connected with the glass transition, is visible on the thermogram around 304 K (Figure 1c). To elucidate this discrepancy, first, we screen the previously reported physicochemical properties of other glycine imino esters and compounds with C=N bond (see Appendix A in Appendix A for more details) [6,8,21,34,35]. In general, the glass transition temperature (*T_g_*) of such systems increases with molar mass (*M*) (Figure 1d) and fulfills the mathematical power law $T_{g}\left(M\right)\propto M^{\alpha }$ with *α* ≈ 0.77. Just like for other non-polymeric compounds, such a power-like relationship among Schiff bases can be explained by increasing energy requirements to induce mutual molecular motions in systems with higher molar mass [36]. This tendency also remains valid among structurally related glycine imino esters, which explains why compound **2** is characterized by higher *T_g_* value than compound **1**.

Apart from this difference, compounds **1** and **2** behave similarly in supercooled liquid and glassy states. This is particularly well reflected in XRD studies. Namely, the X-ray diffraction patterns of the studied melt-quenched compounds (Figure 2a) revealed that they form a supramolecular organization. The diffraction pre-peak appearing at ~0.75 Å^−1^ is an indicator of a medium-range intermolecular order. Actually, the collected diffractograms are very similar to those registered by us for other Schiff bases [21]. Following our previous report [21], the pre-peak’s origin may be ascribed to the organization of molecules in dimeric aggregates stabilized by BHBs. At higher temperatures, the XRD patterns do not change significantly compared to lower temperatures (Figure 2a) for both Schiff bases studied herein. In each case, the intensity of the main peak slightly decreases with the increase in temperature. It is the expected Debye-Waller effect, resulting from an increased thermal energy. In turn, the intensity of the pre-peak slightly increases with the increase in temperature for both Schiff bases. This behavior is not trivial, as the intensity of the pre-peak depends on both the amount and the architecture of the supramolecular clusters. The temperature-induced changes are greater for compound **2**. Nevertheless, they are much smaller than those observed for other H-bonded self-organizing systems, e.g., numerous alcohols [37,38]. It suggests that there are only small temperature-induced changes in the architecture of the supramolecular structure going beyond the standard thermal effect for both Schiff bases in question. Therefore, one may assume that their supramolecular organization scheme is relatively stable at this temperature range.

In order to confirm the hypothesis of self-organization through atypical BHBs, additional FTIR studies supported by DFT calculations were conducted. The most interesting spectral region for compounds **1** and **2** is 2000–3700 cm^−1^ because it constitutes a fingerprint for hydrogen bonds. As presented in Figure 2b,c, numerous bands are distinguishable in this range. Firstly, there are several intense bands between 2800 and 3150 cm^−1^. According to DFT calculations, they can be associated with vibrations of aromatic and aliphatic =C-H, =C-H-, -CH_3_ and -CH_2_- moieties (label as CH_x_ for the sake of simplicity). The second characteristic spectral feature for both compounds is a broad band in between 2400–3500 cm^−1^ of small intensity and subtle structure. Such a band in Schiff bases is usually connected with strong intramolecular hydrogen bonds involved in the formation of an additional six-membered pseudoaromatic chelate ring [15,25]. Indeed, the performed DFT calculations show that formation of such bonds is also energetically privileged for compounds **1** and **2**. Namely, their conformers with intramolecular hydrogen bonds are characterized by significantly smaller total energy than those without such subtle bonds. The differences of 46.80 and 47.28 kJ/mol for compounds **1** and **2**, respectively, demonstrate that intramolecular H-bonds play an essential role in the structural stabilization of both studied Schiff bases. Thanks to them only, the N, H, O atoms and an adjacent phenyl ring are co-planar, which enables the formation of an additional planar six-membered pseudoaromatic chelate ring in the studied compounds. As a consequence, a significant part of their molecular skeleton is stiffened, which, in turn, allows stabilizing the supramolecular clusters. The tendency toward self-organization is additionally enforced by increasing dipole–dipole interaction strength. Namely, the formation of intramolecular hydrogen bonds increases the dipole moment of compounds **1** and **2** from 6.80 and 6.56 D to 7.67 and 7.56 D, respectively. Hence, the performed DFT calculations point that self-organization may be energetically favored. A possible outcome of this process may be the formation of centrosymmetric dimers through BHBs. The FTIR spectra of compounds **1** and **2** seem to confirm this scenario (see Figure 2b,c). They do not contain a broad and intense band, characteristic for H-bonded systems with intermolecular O-H^…^O organization scheme [37,38]. Moreover, there is no intense band above 3600 cm^−1^ stemming from stretching vibrations of free OH groups. Finally, bands arrangement above 3200 cm^−1^ is highly similar for compounds **1** and **2** and other Schiff bases with confirmed self-organization via BHBs [21]. Therefore, it is rational to assume that such a phenomenon also takes place for the studied systems.

Indeed, the performed DFT calculations confirm that the formation of such atypical dimers is possible for compounds **1** and **2** despite their considerable steric hindrance (Figure 2d). Total dipole moment of the dimeric structures is close to 0 D. The intramolecular hydrogen bonds are particularly short. Moreover, all atoms in the BHBs center (including N, H and O) are coplanar with the neighboring phenyl ring, in line with the so-called Parthasarathy rule [39]. Consequently, the six-membered pseudoaromatic chelate ring is highly likely to also be formed in dimeric forms of compounds **1** and **2**.

Each analyzed Schiff base can be simplified by a molecular model that consists of a polar rigid imine unit with attached rotatable elements. Such architecture opens up the possibility to track molecular dynamics on various scales by using broadband dielectric spectroscopy. Herein, we focus only on the imaginary part of the complex dielectric permittivity, *ε*″. Figure 3a,b depicts dielectric loss spectra as a function of frequency, collected for melt-quenched samples at various temperatures. A similar image is observed for both Schiff bases. Namely, two well-separated relaxation processes and a dc-conductivity branch are observed between 133 and 363 K for the probing frequency range 10^−1^–10^6^ Hz. The first dielectric process (labelled as β-process) occurs much below *T_g_*. Its loss peaks are broad, symmetrical and of low intensity, which is typical for secondary relaxations in glass-forming compounds. The relaxation moves toward higher frequencies as the temperature increases and, eventually, becomes too fast to be monitored below 10^6^ Hz. In turn, the maximum of the main process becomes detectable above 10^−1^ Hz at ~271 K for compound **1** and ~311 K for compound **2**, i.e., above *T_g_* (see insets in Figure 3a,b). Noteworthy is also the fact the loss peaks are asymmetrical and followed only by a dc-conductivity branch (Figure 3a,b). All these features are characteristic of the structural α-process that originates from cooperative motions of molecules in the liquid state. Hence, the main α-process steadily shifting toward higher frequencies characterizes an increase in the molecular mobility. An additional hallmark of the dielectric response of both Schiff bases is a kink in the high-frequency slope of the α-relaxation. It is visible around 253–259 K and 295–303 K for compounds **1** and **2**, respectively. Such a peculiarity is usually connected with an intermolecular secondary relaxation in glass formers, named as the Johari–Goldstein (JG) process [3,40]. Unfortunately, the kink becomes quickly covered by the dominating α-relaxation, so the ascription of its origin was omitted.

To complete the discussion on the dielectric loss spectra above *T_g_*, we dissect the loss peak shape. The relaxation anomalies are well described by a single Havriliak–Negami fitting function (HN) [41] with added dc-conductivity contribution:(1)$$\varepsilon^{*}=\frac{\sigma }{\varepsilon_{0}\omega }+\varepsilon_{\infty }+\frac{\Delta \varepsilon }{{\left(1+{\left(i\omega \tau_{HN}\right)}^{\alpha }\right)}^{\beta }}$$
where $\frac{\sigma }{\varepsilon_{0}\omega }$ term determines the conductivity contribution, *ε^*^*—complex permittivity, Δ*ε*—dielectric strength, *ε_∞_*—the high-frequency limit of permittivity, *ω—*angular frequency, *τ—*the relaxation time, *α* and *β*—the shape parameters. The shape parameters differ from the Debye-like dependence (*α* = *β* = 1) and fall into the following ranges: *α* ∈ (0.89; 0.92), *β* ∈ (0.56; 0.62) for compound **1** and *α* ∈ (0.91; 0.94), *β* ∈ (0.57; 0.61) for compound **2**. Moreover, no additional slower Debye mode is detected, which is also manifested in the *α* parameter differing from 1. According to the theory of Déjardin et al., the Debye peak becomes pronounced due to positive cross-correlation between the adjacent molecules (dipole moments) in self-organizing systems [42]. Considering the above finding and the previously discussed diffraction results, the lack of the Debye peak suggests a self-organization of compounds **1** and **2** into agglomerates where the dipole moments cancel out. One can hypothesize that centrosymmetric dimers are formed, similar to those observed in crystal structures of other Schiff bases. In such a case, negative cross-correlation between molecular dipole moments occurs.

The parametrization of the α-process also allowed us to determine the structural relaxation times, *τ_α_*, for compounds **1** and **2**, and thus characterize their molecular dynamics in the liquid phase more precisely. The *τ_α_* values were calculated based on the fitting parameters according to the formula [40]:(2)$$\tau_{\alpha }=\tau_{HN}[sin\left(\frac{\alpha \pi }{2\beta +2}\right){]}^{-\frac{1}{\alpha }}[sin\left(\frac{\alpha \beta \pi }{2\beta +2}\right){]}^{\frac{1}{\alpha }}$$

Relaxation times of the β-process, *τ_β_*, were obtained in a similar way. However, the analysis with a Cole–Cole fitting function [43] (*α* ≠ 1, *β* = 1) was performed due to symmetrical shape of the loss peaks. As presented in Figure 3c, *τ_α_* of both Schiff bases increase while cooling in a non-linear way when plotted versus 1000/T, indicating that molecular mobility becomes slower and activation energy increases when approaching the *T_g_*. The dependences can be well described by a single Vogel–Fulcher–Tamman–Hesse (VFTH) equation:(3)$$\tau_{\alpha }=Aexp\left(\frac{B}{T-T_{0}}\right)$$
where *A* is a pre-exponential factor, *B* is a material constant, and *T_0_* is the so-called ideal glass temperature [44,45,46]. The extrapolation of the fitting VFTH curve up to *τ_α_* = 100 s allows estimating the *T_g_* value. The so-calculated glass transition temperature takes the value of 262 ± 1 for compounds **1** and 302 ± 1 for compounds **2**, in agreement with calorimetric measurements (see Table 1). Based on the extrapolation, one also reaches the conclusion that, surprisingly, temperature causes similar changes in relaxation dynamics for both compounds in the very vicinity of *T_g_* (i.e., for *T_g_*/*T* > 0.95), despite differences in size and stiffness of their molecular scaffold. This feature is apparent on the Angell plot, which showcases the log*τ_α_* = f(*T_g_*/*T*) curve (see inset in Figure 3c) [47]. Consequently, both compounds have similar steepness index values, *m_p_*, defined as the slope of this curve at *T_g_* [3]:(4)$$m_{p}={\left.\frac{dlog_{10}\tau_{\alpha }}{d\left(T_{g}/T\right)}\right|}_{T=T_{g}}$$

Namely, *m_p_* parameter is equal to 82 ± 2 and 83 ± 2 for compounds **1** and **2**. Interestingly, this feature extends to other glass-forming glycine imino esters, as derived from the previously published data (see Appendix A in Appendix A). In other words, *m_p_* basically does not depend on molecular weight and stiffness in this family of Schiff bases, making it different from low-molecular hydrogen-bonded systems.

Contrary to *τ_α_*, *τ_β_* changes linearly with 1000/T, obeying the Arrhenius law:(5)$$\tau_{\beta }=\tau_{0}exp\left(\frac{E_{a}}{RT}\right)$$
where *R* and *E_a_* are gas constant and activation energy, respectively (Figure 3c). The latter parameter is equal to 23 ± 2 kJ∙mol^−1^ for both Schiff bases. Based on this information, one can easily find the $\frac{E_{a}}{RT_{g}}$ ratio equal to 10.6 ± 0.9 and 9.2 ± 0.8 for compounds **1** and **2**, respectively. These values are much lower than 24, which (following the theory of Kudlik et al. [48,49,50,51]) confirms intramolecular conformational changes as the β-relaxation origin. Indeed, conformational analysis performed by the quantum DFT method for the changeable substituent returned comparable energy barriers (*E_g_*) with the experimental ones (c.f., Figure 3d,e with Table 1). However, due to structural complexity, we cannot exclude different mechanisms of conformational changes in both Schiff bases.

To continue the discussion on the structure–dynamics relationships in the glycine imino esters, we carried out extensive high-pressure dielectric measurements. As shown for the representative outcome from isothermal studies of compound **2** at *T* = 351 K, the stepwise pressure increase moves the *α*-relaxation peak toward lower frequencies, although the same thermal energy is delivered to the system (Figure 4a). In other words, compression contributes to the retardation of molecular mobility due to density changes in the liquid phase solely.

The pressure-induced time-scale changes in molecular dynamics are presented as a temperature-pressure surface of *τ_α_* in Figure 4b,c. Ambient pressure alike, *τ_α_* were calculated based on the relaxation peaks parametrization with HN function. The resultant *p*-*T* surfaces of *τ_α_* are not flat for both Schiff bases and obey the Avramov model [52] (see Appendix A for more details). Therefore, the modified Avramov equation was used to obtain the thermal evolution of activation volume, Δ*V* and pressure dependence of *T_g_* for each compound. The first parameter is defined for α-relaxation by the formula:(6)$$\Delta V=RTln\left(10\right){\left(\frac{dlog\tau_{\alpha }}{dp}\right)}_{T}$$

It determines the difference between the volume occupied by molecules in activated and non-activated states [3]. It is also believed to determine the volume requirements for their reorientational motions in a liquid because a strong correlation with molecular weight (*M_w_*) and molar volume (*V_m_*) was empirically found [53,54,55]. As expected, one can also find such interplays among glycine imino esters. Namely, Δ*V* increases with *τ_α_* (i.e., when approaching *T_g_*) for both Schiff bases, always taking higher values for a more bulky compound **2** (see inset in Figure 4d). In other words, the more sizeable a Schiff base is, the more its molecular dynamics is affected by pressure changes (Figure 4d). However, it is surprising to find a simple correlation between Δ*V* and pressure dependence of *T_g_* on a phase diagram. The *T_g_*(*p*) curve was obtained from the modified Avramov model, assuming that the glass transition occurs isochronously at *τ_α_* = 100 s [56]:(7)$$T_{g}\left(p\right)=T_{g\left(0.1MPa\right)}*{\left(1+\frac{p}{\Pi }\right)}^{\beta /F_{0}}$$

In this equation, *Π*, *β* and *F_0_* are free parameters in the Avramov model, and *T_g(0.1 MPa)_* is the glass transition temperature under ambient-pressure conditions. Typically, for glass-forming liquids, the *T_g_* of both Schiff bases rises with pressure in a non-linear way (Figure 4e). Such a tendency can be parametrized with the phenomenological Anderson–Anderson equation:(8)Tg=k11+k2k3P1k2
where *k_1_*, *k_2_*, *k_3_* are material constants [3,57]. The ratio of *k_1_* and *k_3_* determines the pressure coefficient of the glass transition temperature in the limit of low pressures, d*T_g_*/d*p* (*k_1_*/*k_3_* = d*T_g_*/d*p*), which is an important material constant [3]. As a result, we obtained d*T_g_*/d*p* equal to 230 ± 2 and 246 ± 2 K∙GPa^−1^ for compounds **1** and **2**, respectively. Both values are atypically high for self-organizing hydrogen-bonded systems, comparable rather with non-associated van der Waals liquids for which d*T_g_*/d*p* varies within 200–300 K∙GPa^−1^ [58,59]. This exceptional feature may result from a rather low degree of association and/or small size of H-bonded clusters, supporting the hypothesis of self-organization into centrosymmetric dimers via bifurcated hydrogen bonds. More surprising is, however, the tunability of d*T_g_*/d*p*. Namely, this parameter depends on both *m_p_* and Δ*V* variables [3] because
(9)$$m_{p}=\frac{\Delta V}{2.303R\frac{dT_{g}}{dp}}=\frac{\partial log_{10}\tau_{\alpha }}{\partial T_{g}/T}$$

Considering that *m_p_* is basically constant for medium-sized Schiff bases from the family of glycine imino esters, d*T_g_*/d*p* is directly proportional to Δ*V*, becoming strictly correlated with molecule size. Thus, one can expect to achieve higher *T_g_* and d*T_g_*/d*p* values by increasing the molar mass and molar volume of a glycine imino ester, respectively. These simple relations explain why higher *T_g_* and d*T_g_*/d*p* values characterize compound **2**.

Atypically high values of d*T_g_*/d*p* coefficient force us to pose the question of whether hydrogen bonds are really the main interactions determining molecular dynamics and dielectric properties of glycine imino esters. In order to deliver an answer to this issue, we turn back to the ambient-pressure dielectric results. Based on them, we examine the fundamental relation between dielectric increment assessed near *T_g_* (Δ*ε*(*T_g_*)) and the frequency dispersion width of the α-relaxation loss peak, quantified by the fractional exponent in the Kohlrausch–Williams–Watts function (i.e., *β_KWW_* parameter) [60,61]. According to this empirical relationship, dielectric loss peak becomes narrower and *β_KWW_* rises when dielectric increment increases [61]. This peculiarity, apparent on the semilogarithmic plot of *kT_g_*(Δ*ε*(*T_g_*))^2^ versus *β_KWW_*, remains valid only for polar van der Waals systems and fails mainly for nonpolar compounds or H-bonded liquids, e.g., alcohols (see Figure 5) [61,62]. Disagreement between Δ*ε* and *β_KWW_* among alcohols has been shown only for 4-phenyl-2-butanol, 4-phenyl-1-butanol and 1-phenyl-2-butanol so far [62].

The herein-studied compounds **1** and **2**, as well as the previously reported glycine imino esters are polar, which makes the empirical relationship useful to determine the significance of hydrogen bonds in these systems. As presented in Figure 5, the near-*T_g_* values of Δ*ε* and *β_KWW_* (obtained from the Δ*ε*, *α*, *β* parameters of HN fit function, as $\beta_{KWW}=\alpha \cdot \beta$) can be rationalized by this correlation in the case of glycine imino ester family. It means that these compounds behave like simple van der Waals liquids. In other words, despite the occurrence of intra- and intermolecular hydrogen bonds, weak van der Waals and dipole–dipole interactions have a decisive impact on their molecular dynamics and dielectric properties of glycine imino esters. This conclusion coincides with atypically high d*T_g_*/d*p* values for compounds **1** and **2**. It also supports the hypothesis of self-organization into dimeric structures via BHBs because, in such case, the hydrogen bonding sites are ‘used up’ for the formation of associates. Hence, the performed dielectric and X-ray diffraction studies show that hydrogen bonds are responsible for stiffening of the molecular skeleton and intermolecular self-organization, whereas van der Waals and dipole–dipole interactions control the molecular dynamics and dielectric properties of the medium-sized compounds **1**, **2** and other glycine imino esters. The obtained result also brings us to a general conclusion that (contrary to popular belief) compounds with OH groups may behave like simple van der Waals liquids, particularly when the moieties are involved in the formation of strong intramolecular hydrogen bonds and bifurcated hydrogen bonds.

## 3. Materials and Methods

### 3.1. Materials

The herein-studied compounds **1** and **2** are medium-sized and sterically hindered model representatives of Schiff bases belonging to the subgroup of glycine imino esters. They are commercially unavailable, and their synthesis with spectral analysis and purity confirmation were recently reported by us [33].

### 3.2. Differential Scanning Calorimetry

Calorimetric measurements of compounds **1** and **2** were conducted on a Mettler-Toledo DSC apparatus (Mettler-Toledo, Greifensee, Switzerland) equipped with a liquid nitrogen cooling accessory and a HSS8 ceramic heat sensor with 120 thermocouples. Samples were measured in sealed aluminum pans with a volume of 40 μL. An empty sealed aluminum pan of the same volume was used as a reference sample. The thermograms were collected during heating scans between 210 and 350 K. The heating rate was equal to 10 K∙min^−1^. All measurements were performed in the atmosphere of nitrogen, the flow of which was set to 60 mL∙min^−1^.

### 3.3. X-ray Diffraction Studies

X-ray diffraction measurements of Schiff bases **1** and **2** were performed on a Rigaku-Denki D/MAX RAPID II-R diffractometer (Rigaku Corporation, Tokyo, Japan) equipped with a rotating Ag anode, an incident beam (002) graphite monochromator and an image plate in the Debye−Scherrer geometry. Samples were measured in glass capillaries with a diameter of 1.5 mm, and the temperature was controlled by an Oxford Cryostream Plus and Compact Cooler. The collected two-dimensional diffraction patterns were converted into one-dimensional intensity data versus the scattering vector:(10)$$Q=4\pi \left(sin\theta \right)/\lambda$$
where 2θ is the scattering angle, and the wavelength of the incident beam, λ*,* is equal to 0.56 Å.

### 3.4. Fourier Transform Infrared Spectroscopy

Fourier transform infrared studies of Schiff bases **1** and **2** were performed at room temperature. A Thermo Scientific IS50 spectrometer (Thermo Fisher Scientific, Madison, WI, USA) equipped with a standard source and DTGS Peltier-cooled detector was utilized for them. Spectra were recorded at an absorbance mode between 400 and 4000 cm^−1^ with a spectral resolution of 4 cm^−1^. The accumulation of 16 scans was chosen as the experimental condition. Measurements were conducted on melted samples, placed between two CaF_2_ glasses. The obtained data were subjected to the baseline, water and carbon dioxide correction.

### 3.5. Broadband Dielectric Spectroscopy under Ambient Pressure

Ambient-pressure molecular dynamics of compounds **1** and **2** was studied by means of a broadband dielectric spectroscopy. Measurements were conducted on a stainless-steel capacitor of 10 mm diameter, sealed with a Telfon ring and completely filled with the studied melted material. The distance between its parallel plates was set to 100 μm and kept by two quartz spacers. The dielectric spectra were registered by means of Novocontrol Broadband Dielectric Spectrometer (NOVOCONTROL Technologies GmbH & Co. KG, Montabaur, Germany) equipped with the Alpha Impedance analyzer. They were collected in the frequency range of 10^−1^–10^6^ Hz in a wide temperature range of 133–311 K (for compound **1**) or 133–363 K (for compound **2**) with steps Δ*T* = 5 K below the glass transition temperature (*T_g_*) or Δ*T* = 2 K for temperatures in the vicinity and above *T_g_*. Temperature was controlled during measurements by a Novocontrol Quattro system and stabilized with use of nitrogen gas with precision better than 0.2 K. The dielectric spectra were analyzed in the representation of the complex dielectric permittivity:*ε^*^*(*f*,*T*) *= ε*′(*f*,*T*) – i*ε*″(*f*,*T*), (11)
where *ε*′ and *ε*″ are its real and imaginary parts, respectively. The analysis procedure was performed in the WinFit program (NOVOCONTROL Technologies GmbH & Co. KG, Montabaur, Germany).

### 3.6. Broadband Dielectric Spetroscopy under High-Pressure Conditions

Dielectric experiments under elevated pressure were performed on high-pressure system delivered by Unipress, Warsaw, Poland. Its most critical elements are a high-pressure chamber (made of beryllium bronze) equipped with a thermostatic mantle and a high-pressure closure with electric connections, preliminary hand pump and an automatic micropump of the MP5 type with a pressure controller. Measurements were conducted on a stainless-steel capacitor of 10 mm diameter, the parallel plates of which were distanced by a Teflon spacer. The capacitor (completely filled with a melted material) was sealed and covered by a Teflon tape to separate it from the high-pressure medium (silicon oil of HL 80 type). During the measurements, the pressure was controlled by a Honeywell tensometric meter with a precision of 1 MPa. In turn, temperature was adjusted and controlled by Julabo Presto thermostatic bath with a precision of 0.2 K. The dielectric spectra were registered in between 10^−1^–10^6^ Hz by means of the Novocontrol Broadband Dielectric Spectrometer (NOVOCONTROL Technologies GmbH & Co. KG, Montabaur, Germany) equipped with the Alpha Impedance analyzer.

### 3.7. Density Functional Theory Calculations

All DFT calculations using an efficient hybrid density functional B3LYP were performed with Gaussian 16 program suite [63,64,65,66,67,68]. Full geometry optimization and IR spectra simulation were conducted with 6-311+G* basis set for monomeric forms. In turn, the geometries of their dimers were predicted using a smaller 6-31G* basis set. All calculations were performed in the gas phase. The luck of negative harmonic frequencies indicated the optimized geometries corresponded to the minimum of potential energy for the investigated molecules. Subsequently, conformational analysis was performed for both compounds in question. During these calculations, angular alteration for the C4-C3-C2-C1 dihedral angle was performed for the alkyl -C_4_H_9_ substituent in compound **1** (Figure 1a). These calculations were performed from -179^0^ to 191^0^ by 10^0^. On the other hand, the -CH=CH-Ph substituent of compound **2** was rotated in the range from -114^0^ to 256^0^ by 10^0^ around the C10-C9-C8-C7 dihedral angle (as marked in Figure 1a). The corresponding energy changes for those substituent rotations were shown in the corresponding potential energy curve.

## 4. Conclusions

The general rules that govern the ambient- and high-pressure molecular dynamics and near-glass transition phenomena are derived for Schiff bases belonging to the family of glycine imino esters. Firstly, atypical self-organization via bifurcated hydrogen bonds into centrosymmetric dimers is proven based on X-ray diffraction, infrared and dielectric studies. Although these hydrogen bonds are relatively weak, the associates are formed even at high temperatures (T > 400 K). Secondly, the tunability of the glass transition temperature, *T_g_*, and its pressure coefficient, d*T_g_*/d*p*, are observed for this Schiff base family. One can expect to achieve higher *T_g_* and d*T_g_*/d*p* values by increasing the molar mass and molar volume of a glycine imino ester, respectively. The first relationship can be described by a general power law: $T_{g}\left(M\right)\propto M^{\alpha }$, with *α* being close to 0.77. Independently of *T_g_*, the molecular dynamics changes in a similar way when approaching the glass transition at ambient pressure. Consequently, surprising tunability of d*T_g_*/d*p* parameter with a simple rule d*T_g_*/d*p* ∝ Δ*V* is observed among the glycine imino esters. Excellent thermal stability, as well as atypically high and tunable *T_g_*, d*T_g_*/d*p* values, are the most distinguishable features of this Schiff base family. Finally, this manuscript proves that despite the presence of intra- and intermolecular hydrogen bonds, weak van der Waals and dipole–dipole interactions are the main forces governing the molecular dynamics and dielectric properties of glycine imino esters. Consequently, the obtained result brings us to a general conclusion that (contrary to popular belief) compounds with OH groups may behave like simple van der Waals liquids, particularly when the moieties are involved in the formation of strong intramolecular hydrogen bonds and bifurcated hydrogen bonds.

## Figures and Tables

**Figure 1 ijms-23-05185-f001:**
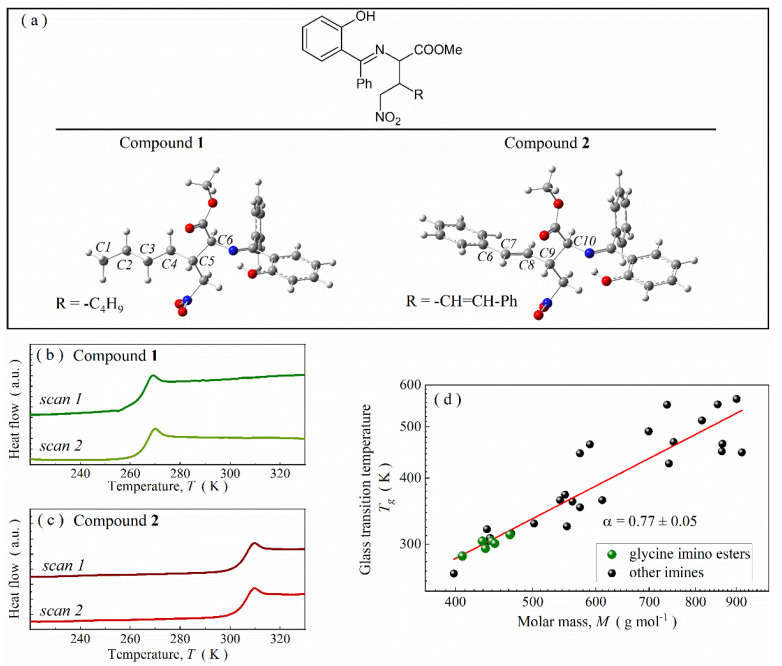
(**a**) The structure of glycine imino ester molecules of particular interest with labeled most crucial carbon atoms for conformational analysis. (**b**) Outcome of two subsequent calorimetric heating scans for compound **1**. (**c**) Thermograms for compound **2**. (**d**) Correlation between molar mass and glass transition temperature for imines.

**Figure 2 ijms-23-05185-f002:**
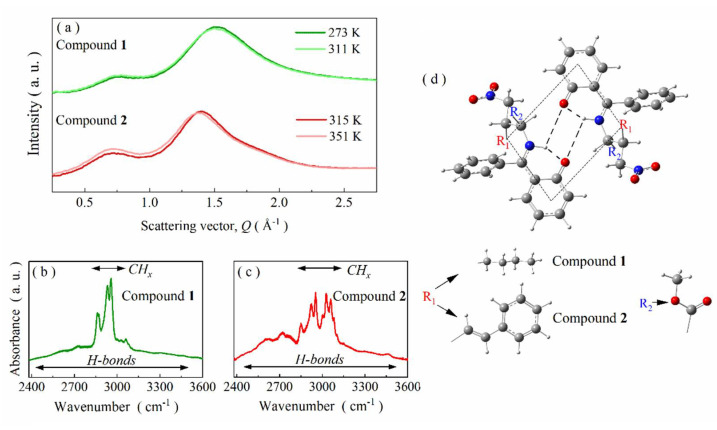
(**a**) XRD patterns collected for melt-quenched glycine imino esters **1** and **2** at various temperatures. (**b**) Room-temperature FTIR spectrum of compound **1** in the H-bonding region. (**c**) Room-temperature infrared spectrum of compound **2** presented from 2400 to 3600 cm^−1^. (**d**) DFT-optimized dimeric structure with marked BHBs center.

**Figure 3 ijms-23-05185-f003:**
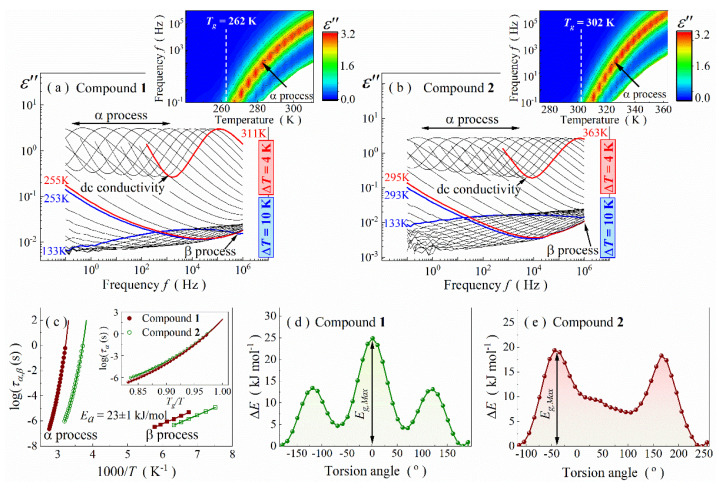
Dielectric loss spectra of compounds **1** (**a**) and **2** (**b**). Insets show temperature evolution of their α-relaxation peak. (**c**) Relaxation map for both Schiff bases with inserted Angell plot. (**d**) Rotational curve inside alkyl -C_4_H_9_ substituent (C4-C3-C2-C1 dihedral angular alteration, i.e., around C5-C4 bond) of compound **1.** (**e**) Rotational curve for -CH=CH-Ph substituent (C10-C9-C8-C7 dihedral angular alteration, i.e., around C9-C8 bond) in compound **2**.

**Figure 4 ijms-23-05185-f004:**
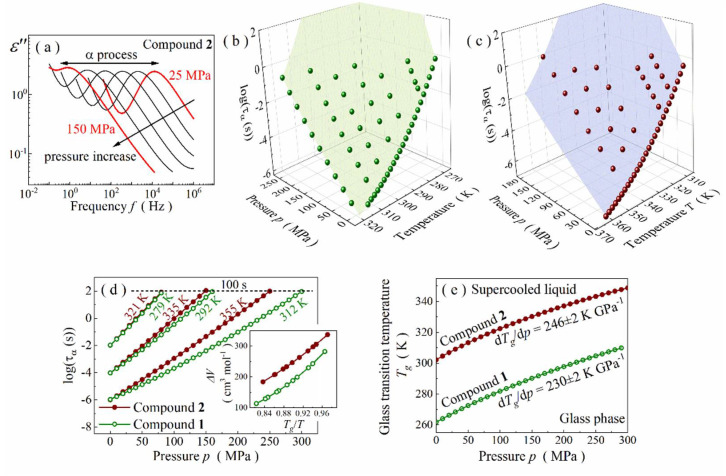
(**a**) Representative spectra measured at isothermal conditions of 351 K for compound **2**. (**b**) Pressure-temperature surface of *τ_α_* for compound **1**. (**c**) Pressure-temperature dependence of *τ_α_* for compound **2**. (**d**) Comparison of pressure-induced changes in relaxation times between compounds **1** and **2**. Insets show temperature variation of their activation volume values. (**e**) Phase diagram for compounds **1** and **2**.

**Figure 5 ijms-23-05185-f005:**
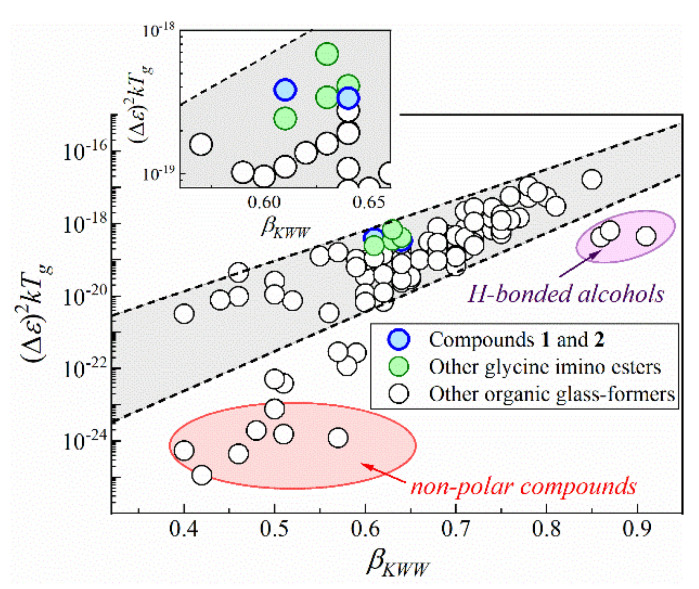
Fundamental correlation between dielectric strength Δ*ε* and the frequency dispersion width of the α-relaxation loss peak demonstrated by plotting *kT_g_*(Δ*ε*(*T_g_*))^2^ against *β_KWW_*.

**Table 1 ijms-23-05185-t001:** Molar mass (*M*), glass transition temperature (*T_g_*) calculated from DSC and BDS experiments, ambient-pressure fragility index (*m_p_*), activation energy for secondary relaxation (*E_a_*), energy barrier for rotation of the changeable substituent (*E_g_*) and pressure coefficient of glass transition temperature (d*T_g_*/d*p*) for compounds **1** and **2**.

Parameter	Compound 1	Compound 2
*M* (g∙mol−1)	398	444
*T_g_* from DSC (K)	~263	~304
*T_g_* from BDS (K)	262 ± 1	302 ± 1
*m_p_*	82 ± 2	83 ± 2
*E_a_* (kJ∙mol−1)	23 ± 1	23 ± 1
*E_g_* (kJ∙mol−1)	25	20
d*T_g_*/d*p* (K∙GPa−1)	230 ± 2	246 ± 2

## Data Availability

The data presented in this study are available on request from the corresponding author.

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
