# Peer review of "Simple Rules for Complex Near-Glass-Transition Phenomena in Medium-Sized Schiff Bases"

_ijms, 2022, doi:10.3390/ijms23095185_

Round 1

Reviewer 1 Report

The authors have reported and discussed several mathematical equations describing physicochemical properties of  Schiff bases with particular consideration of the relationship between their structure and dynamics. Although adequate discussion have been provided, the article requires thorough revision to minimize some editorial errors. I believe that the original subject matter and the high value of the content makes the publication suitable for printing in its present form.

Below are some specific comments:

I suggest adding a list of abbreviations used at the beginning of the article, which will make reading and interpreting the text much easier.

Table 1

Please move units to the first column “Parameter”.

Figure 4a and 4b

Units and axis descriptions are unreadable, please improve chart quality/axis angle.

1.2. Differential scanning calorimetry

What was the reference sample, an empty pan?

Author Response

Reviewer: The authors have reported and discussed several mathematical equations describing physicochemical properties of  Schiff bases with particular consideration of the relationship between their structure and dynamics. Although adequate discussion have been provided, the article requires thorough revision to minimize some editorial errors. I believe that the original subject matter and the high value of the content makes the publication suitable for printing in its present form.

Answer: We are very thankful for the review and valuable comments which helped us improve the manuscript. The corrections were made according to all the Reviewer’s comments. Please note that all revisions made to our manuscript are marked up using the “Track Changes” function.

Comment 1

Reviewer: I suggest adding a list of abbreviations used at the beginning of the article, which will make reading and interpreting the text much easier.

Answer: We are thankful for this remark. We added a list of the most frequently used abbreviations to the introduction. The following text was added at the end of this part of our manuscript: In this article, we frequently use some symbols and abbreviations apart from those connected with the names of experimental techniques. Therefore, for the sake of simplicity, we list them in an alphabetical order and provide their explanation: α-relaxation- structural relaxation, βKWW- fractional exponent in the Kohlrausch-Williams-Watts function, Δε-dielectric increment, ΔV- activation volume, ε″- imaginary part of complex dielectric permittivity, τα- structural relaxation time, τβ- relaxation time of the β-process, BHBs- bifurcated hydrogen bonds, dc-conductivity- direct current conductivity, dTg/dp-pressure coefficient of the glass transition temperature, Ea- activation energy, Eg- calculated energy barrier for conformational change, H bond- hydrogen bond, HN function- Havriliak-Negami function, JG process- Johari-Goldstein process, k- Boltzmann constant, M-molar mass, mp- steepness index, p- pressure, T- temperature, Tg- glass transition temperature, VFTH equation- Vogel-Fulcher-Tamman-Hesse equation.’

In the introduction, we also explained abbreviations of the experimental techniques utilized by us, which was omitted mistakenly in the previous version of the manuscript.

Comment 2

Reviewer:  Table 1  Please move units to the first column “Parameter”.

Answer: Units were moved to the column “Parameter” in Table 1.

Comment 3

Reviewer:  Figure 4a and 4b  Units and axis descriptions are unreadable, please improve chart quality/axis angle.

Answer: The Reviewer was speaking probably about Figure 4b and 4c, in which the axis angle is changeable. We are thankful for this remark. We improved the quality of whole graph 4, changed the axis angles in Figure 4b and 4c. We also noticed that quality of insert in Figure 4d was poor. Therefore, we modified slightly this chart.

Comment 4

Reviewer:  Differential scanning calorimetry  What was the reference sample, an empty pan?

Answer: We are thankful for this question. We added the following information to the manuscript: ‘An empty sealed aluminum pan of the same volume was used as a reference sample.

Reviewer 2 Report

The Authors of this manuscript measure and analyze temperature- and pressure-dependent dielectric spectra of two glycine imino esters, and compare their results to recently published results by the same authors on similar compounds [J. Molecular Liquids 348, 118052].

The characterization is well-carried out, and the fitting procedures and data analysis is sound. However, the main conclusions of the manuscript need to be further corroborated or discussed. Two main conclusions of the paper should be further analyzed and correlated with one another are the existence and persistence of dimers, and the role of hydrogen bonding. In detail:

1) I agree with the Authors that XRD provides evidence for molecular clustering, but can they provide spectroscopic evidence (NMR, IR) for the formation or the geometry of dimers? (they provide IR characterization of related compounds in their previous work) Could the dimers actually involve more conventional hydrogen bonding? Could clustering occur by formation of a single H-bond between two molecules? Perhaps DFT comparisons of different geometries/H-bonding motifs might be helpful. Or also comparison with related data on previously studied compounds.

2) Related to this, could the Authors analyze the temperature- or pressure- stability of these dimers? Looking at XRD results, it seems that at least for compound 2, increasing the temperature might reduce the amount of dimers present, as the prepeak has a relatively lower intensity at higher T (something that could be expected as a result of increased therma energy). Do the Authors agree? Is there any evidence for temperature- or pressure- dependent variation of the dielectric spectra (HN exponents, strength, Kirkwood factor) that could be ascribed to variation in the clustering/population of dimers? If not, can the Authors provide evidence for the stability of the dimers for different P and T conditions? Are there previous studies on this aspect of Schiff bases?

3) In figure 1(d), Authors show the correlation between molecular mass and Tg. If the dimers are stable and relatively rigid, the relaxing unit is the dimer, which has twice the mass of the monomer. Could the Authors compare in the same figure the Tg values obtained with those of molecules of similar mass, but that do not form dimers, or to molecules that have twice the mass?

4) If the hydrogen bonding sites of the studied molecules are "used up" for the formation of stable dimers, this would rationalize the fact that intermolecular interactions between relaxing units (dimers) appear to be mainly of van der Waals nature. I suppose that this is what the Author mean when the write, at the end of the Results section, that "compounds with OH groups may behave like simple van der Waals liquids, particularly when the moieties are involved into formation of strong intramolecular hydrogen bonds." Few lines before, however the Authors state that this conclusion about the studied compounds "is also supported by their chemical structure, which is dominated by nonpolar substituents sterically hindering access to the N…H…O group". Apart from the fact that the N···H···O group is ill-defined, I think that this sentence is not clear (and seems in conflict with the idea of dimer formation).

If the Authors cannot provide further evidence for the geometry, hydrogen bonding motif, and stability of dimers, they should perhaps only speak more loosely of the formation of H-bonded clusters that interact with one another via vdW forces.

Minor details:

- Authors should better specify what they mean by N···H···O bonds. I suppose it should be O-H···N bonds.

- In Figure 5, they should indicate which markers correspond to H-bonded systems such as alcohols.

- I don't understand the word "sizeable" as employed by the authors. I don't think that the molecules studied are particularly large, and in general, I would prefer a phrasing referring to the mass of the molecules, or something like "medium-size compounds".

Author Response

Reviewer: The Authors of this manuscript measure and analyze temperature- and pressure-dependent dielectric spectra of two glycine imino esters, and compare their results to recently published results by the same authors on similar compounds [J. Molecular Liquids 348, 118052].

The characterization is well-carried out, and the fitting procedures and data analysis is sound. However, the main conclusions of the manuscript need to be further corroborated or discussed. Two main conclusions of the paper should be further analyzed and correlated with one another are the existence and persistence of dimers, and the role of hydrogen bonding.

Answer: We are very thankful for the review and valuable comments which helped us improve the manuscript. Please note that all revisions made to our manuscript are marked up using the “Track Changes” function.

Comment 1

Reviewer: I agree with the Authors that XRD provides evidence for molecular clustering, but can they provide spectroscopic evidence (NMR, IR) for the formation or the geometry of dimers? (they provide IR characterization of related compounds in their previous work) Could the dimers actually involve more conventional hydrogen bonding? Could clustering occur by formation of a single H-bond between two molecules? Perhaps DFT comparisons of different geometries/H-bonding motifs might be helpful. Or also comparison with related data on previously studied compounds.

Answer: We are very thankful for this comment. We made additional FTIR measurements at room temperature and DFT calculations. We also compared the spectra with those reported previously for structural analogues of compounds 1 and 2. The infrared spectra are added to Figure 2 (See Fig. 2b and 2). As can be seen, there is no characteristic broad and intense band stemming from intermolecular hydrogen bonds with O-HO motif (like in alcohols). Therefore the dimers involving more conventional hydrogen bonding can be definitely excluded. The most characteristic feature of the FTIR spectra is a broad band of low intensity and subtle structure, connected with strong intramolecular hydrogen bonds between OH and C=N groups. Actually, the spectra are highly similar for those obtained for other glycine imino esters, for which dimeric structures with bifurcated hydrogen bonds were confirmed. Therefore, based on the performed IR and XRD studies one can definitely say that self-organization processes in compounds 1 and 2 lead to formation of dimers through bifurcated hydrogen bonds. The architecture of the dimeric structures should be centrosymmetric so that the dipole moment is close to 0D. Otherwise, we would observe an additional Debye peak on the dielectric loss spectra. The discussion is also supported by additional DFT measurements in the new version of manuscript. Namely, we calculated two geometries of compounds 1 and 2: with and without intramolecular hydrogen bond. We observed that the total energy of the conformers with intramolecular hydrogen bonds are characterized by significantly smaller total energy than those without such subtle bonds. The differences of 46.80 and 47.28 kJ/mol for compounds 1 and 2, respectively, demonstrate that intramolecular H-bonds play an essential role in structural stabilization of both studied Schiff bases. Moreover, we observed that formation of intramolecular hydrogen bonds increases the dipole moment of compounds 1 and 2 from 6.80 and 6.56 D to 7.67 and 7.56 D. Since the most polar part of compounds 1 and 2 is connected with the C=NH-O moiety, the performed DFT calculations point that self-organization through BHBs may be energetically favoured. New Figure 2 and additional discussion is placed in pages 5 and 6 of the revised version of our manuscript.

Comment 2

Reviewer: Related to this, could the Authors analyze the temperature- or pressure- stability of these dimers? Looking at XRD results, it seems that at least for compound 2, increasing the temperature might reduce the amount of dimers present, as the prepeak has a relatively lower intensity at higher T (something that could be expected as a result of increased therma energy). Do the Authors agree? Is there any evidence for temperature- or pressure- dependent variation of the dielectric spectra (HN exponents, strength, Kirkwood factor) that could be ascribed to variation in the clustering/population of dimers? If not, can the Authors provide evidence for the stability of the dimers for different P and T conditions? Are there previous studies on this aspect of Schiff bases?

Answer: The Reviewer would be generally right. However, we noticed that in the case of Figure 2a, for compound 2 the legend was incorrect – the temperature markings were reversed. We are very sorry for this mistake. The corrected version of Figure 2 is place in page 5 of the revised manuscript. This means that, as in the case of compound 1, the intensity of the pre-peak slightly increases with increase of the temperature. The intensity of the main peak slightly decreases with the increase of temperature – it is an expected effect, as a result of increased thermal energy. In turn, the behaviour of the pre-peak is not trivial and suggests that changes in temperature induce a variation in the architecture of the supramolecular structure going beyond the standard thermal effect. The intensity of the pre-peak depends on both: the amount and the architecture of the supramolecular clusters. The mutual orientation of molecules probably slightly changes as a function of temperature. We are not able to establish in detail the stability (the amount) of these clusters based on XRD and FTIR spectra. However since the pre-peak is present at higher temperatures with very similar intensity and shape, we can assume that the dimeric architecture of the clusters is relatively stable.

We admit, that the wording “one may assume that their supramolecular organization is thermally stable at this temperature range” could be deceptive. Our intention was no to discuss thermal stability in terms of the amount of the dimers, but we would like to point that the dimeric architecture of the clusters does not change with temperature. Therefore, we corrected the sentence to “Therefore, one may assume that their supramolecular organization scheme is relatively stable at this temperature range.” The corrected sentence and additional discussion on temperature-induced changes in XRD spectra of compounds 1 and 2 is placed on page 4 of the revised manuscript.

Comment 3

Reviewer: In figure 1(d), Authors show the correlation between molecular mass and Tg. If the dimers are stable and relatively rigid, the relaxing unit is the dimer, which has twice the mass of the monomer. Could the Authors compare in the same figure the Tg values obtained with those of molecules of similar mass, but that do not form dimers, or to molecules that have twice the mass?

Answer: Such a comparison would be interesting. Unfortunately, we cannot make it. Figure 1(d) depicts the correlation between molecular mass and Tg for glycine imino esters and other previously reported  compounds with C=N bond. The literature compounds that belong to the second group were well characterized by calorimetry, based on which the Tg values were determined (see Refs. 6,8,34,35 from manuscript). However, they were not investigated by XRD or broadband dieletric spectroscopy in the liquid phase or glass state. As a consequence, there is no information about self-organization of these compounds in the literature (Refs. 6,8,34,35 from manuscript). That is why such comparison is impossible to be prepared. In the literature, known is a general relationship between Tg and molar mass for low-weight compounds, which is of the same type as we presented for Schiff bases (see Ref. Polymer 2013, 54, 6987-6991).

Comment 4

Reviewer: If the hydrogen bonding sites of the studied molecules are "used up" for the formation of stable dimers, this would rationalize the fact that intermolecular interactions between relaxing units (dimers) appear to be mainly of van der Waals nature. I suppose that this is what the Author mean when the write, at the end of the Results section, that "compounds with OH groups may behave like simple van der Waals liquids, particularly when the moieties are involved into formation of strong intramolecular hydrogen bonds." Few lines before, however the Authors state that this conclusion about the studied compounds "is also supported by their chemical structure, which is dominated by nonpolar substituents sterically hindering access to the N…H…O group". Apart from the fact that the N···H···O group is ill-defined, I think that this sentence is not clear (and seems in conflict with the idea of dimer formation).

Answer: We are thankful for this remark. We agree that this sentence was misleading. Therefore, we removed it from the manuscript and corrected the whole fragment devoted to intermolecular interactions. We changed it to: “In other words, despite occurrence of intra- and intermolecular hydrogen bonds, weak van der Waals and dipole-dipole interactions have decisive impact on their molecular dynamics and dielectric properties of glycine imino esters. This conclusion coincides with atypically high dTg/dp values for compounds 1 and 2. It also supports the hypothesis of self-organization into dimeric structures via BHBs because in such case the hydrogen bonding sites are ‘used up’ for the formation of associates.

Comment 5

Reviewer: If the Authors cannot provide further evidence for the geometry, hydrogen bonding motif, and stability of dimers, they should perhaps only speak more loosely of the formation of H-bonded clusters that interact with one another via vdW forces.

Answer: We added additional evidence for the hydrogen bonding motif (dimers) based on FTIR spectroscopy and further DFT calculations so “more loosely speaking of the formation of H-bonded clusters that interact with one another via vdW forces” is not needed.

Comment 6

Reviewer: Authors should better specify what they mean by N···H···O bonds. I suppose it should be O-H···N bonds.

Answer: For some Schiff bases, an environment-dependent and resonance-assisted proton-transfer equilibrium may occur in the intramolecular hydrogen bond. As a consequence, there is equilibrium between O-HN and OH-N schemes and thus one can speak about NHO as a resonance hybrid of these two bonding schemes. We wrote about this in our manuscript. Namely, we included the following information: “Namely, an additional planar six-membered pseudoaromatic chelate ring with possible proton transfer in the intramolecular HB is usually formed in systems of internal architecture similar to compounds 1 and 2 [25].” However, we admit that we have no proof that such a phenomenon occurs also in compounds 1 and 2, analyzed by us in this article. Therefore we changed the wording “intramolecular NHO bonds” to “intramolecular hydrogen bonds”. We are very thankful for the comment.

Comment 7

Reviewer: In Figure 5, they should indicate which markers correspond to H-bonded systems such as alcohols.

Answer: We are thankful for this remark. We would like to point that we mistakenly missed the points corresponding to H-bonded systems (alcohols) in the previous version of Figure 5. Therefore, we added them to this graph and marked in purple. We also added the following information to the manuscript: “Disagreement between Δε and βKWW among alcohols has been shown only for 4-phenyl-2-butanol, 4-phenyl-1-butanol and 1-phenyl-2-butanol so far [60].

Comment 8

Reviewer: I don't understand the word "sizeable" as employed by the authors. I don't think that the molecules studied are particularly large, and in general, I would prefer a phrasing referring to the mass of the molecules, or something like "medium-size compounds".

Answer: We are thankful for this remark. We fully agree with the Reviewer and we replaced the word ‘sizeable’ with ‘medium-sized’ in the whole manuscript (including the title).